# Antibiotic Culture: A History of Antibiotic Use in the Second Half of the 20th and Early 21st Century in the People’s Republic of China

**DOI:** 10.3390/antibiotics12030510

**Published:** 2023-03-03

**Authors:** Xun Zhou

**Affiliations:** History Department, University of Essex, Colchester CO4 3SQ, UK; xzhoug@essex.ac.uk

**Keywords:** history of antibiotic development, culture of drug consumption, self-medication, politics of health, antibiotic overuse, marketization of health, health care access, animal welfare, China

## Abstract

Antimicrobial resistance is now widely regarded as a global public health threat. A growing number of studies suggest that antibiotic resistance is higher in China than in most western countries. Despite the current official regulation prohibiting pharmacies from the unrestricted selling of antibiotics, there is little sign of declining consumer demand. China now ranks as the second largest consumer of antibiotics in the world, after India. Drawing on published historical data, unpublished archival documents, and recently collected oral interviews, this paper provides a historical overview of antibiotic use and abuse in the People’s Republic of China (PRC) from the second half of the 20th century to the present. It demonstrates how the political demand for health improvement, along with the state-sponsored popularization of allopathic medicine, on the one hand, and the lack of access to adequate medical care for the majority of the population, as well as the existing culture of self-medication, on the other hand, are working in tandem to create antibiotic dependency in China. In addition, the privatization and marketization of biomedicine and health care in post-Mao China have helped to build a new and ever-thriving network of production, distribution, and marketing of antibiotics, which has often proven difficult for the authorities to monitor. At the same time, increased purchasing power and easier accessibility created by this new network of production, distribution, and marketing have further contributed to the prevalence of antibiotic overuse in the late 20th and early 21st centuries.

## 1. Introduction

The discovery of penicillin by Alexander Fleming and the subsequent development of antibiotics is widely viewed as one of the most important achievements of modern medical science. In the second half of the 20th century, in the excitement surrounding the success of treating a range of previously incurable diseases and serious infections, a broad range of antibiotics was made readily available to the general public, first in the West and then globally. They have helped to save millions of lives. Yet, the powerful “wonder cure” of antibiotics would lead to equally powerful drug resistance. According to the World Health Organization (WHO), antimicrobial resistance (AMR) is now “one of the top 10 global public health threats facing humanity” and “misuse and overuse of antimicrobials are the main drivers in the development of drug-resistant pathogens” [1]. The People’s Republic of China (PRC), being one of the world’s largest producers and the second largest consumer of antibiotics, is also a major contributor to overall global increases in AMR [2,3]. The main objective of this paper is to provide a historical overview of antibiotic use and abuse in the PRC from the second half of the 20th century to the present and to identify some key political, social, cultural, and economic factors that have contributed to AMR in the PRC.

## 2. Methodology

This study draws on published data representing different regions of China, as well as unpublished archival documents and recently collected oral interviews from eight provinces across three regions of China: three in the east (Shandong, Shanghai, Zhejiang), three in the southwest (Sichuan, Yunnan, and Guizhou) and two in the northwest (Gansu and Shaanxi). These eight provinces represent a wide range of socio-economic development in China, and interviewees include experts from medical universities, local cadres, rural primary health professionals, and villagers, although the majority are from rural and economically less developed regions. Interviewees were recruited with the help of existing contacts from regional Centers for Disease Control (CDCs), as well as from contacts I have built up over the past years while researching the social history of the Great Leap Forward famine. For this study, 30 interviews were selected for analysis from over 200 interviews conducted between 2014 and 2019. Interviews collected by the Institute of Health at the Kunming Medical University and Yunnan Health and Development Research Association (YHDRA) were also analyzed. These interviews were conducted by a team at the YHDRA between 2000 and 2002, and are made available in Zhang Kaining, Wen Yiqun, and Liang Ping, eds., *From Barefoot Doctors to Village Doctors* (Kunming: Yunnan Renmin chubanshe 2002).

For this study, techniques were employed, familiar to social historians of medicine, that describe the qualitative stories behind the quantitative statistics. At the same time, transnational and translocal approaches were applied to the evidence to draw out the network of relationships that tie personal and community stories to important issues in global health. Conventional top-down analysis, which focuses on policy formulations, has often failed to provide critical insights into the experiences of ordinary antibiotic consumers. Instead, this research examines previously unstudied local archival data and personal testimonies using statistical analysis and national policies to get a more nuanced understanding of the many complexities involved, that contributed to the prevalence of antibiotic overuse in the late 20th and early 21st century PRC.

## 3. Penicillin as a Gift to War-Torn China

The beginning of the Chinese love affair with antibiotics was from 1944 to 1949. In the dark and stormy summer days of 1944, two months into Operation Ichi-Go, known as the largest military campaign launched by imperial Japan during World War II, the Vice President of the United States, Henry Agard Wallace, brought penicillin to China. It was presented to Generalissimo Jiang Kaishek of China as a state gift from the United States; not merely as a new pharmaceutical [4]. This gift gave hope to a country raging with war and epidemics of cholera, plague, and typhus fever, as well as to the urban public, who were traumatized by both. Penicillin also entered China when opium smoking was on its way out, at least in urban centers such as Shanghai, where it was frowned on by modern, westernized urban Chinese elites as “lamentably old-fashioned”: “Opium was decadent. Opium was for grandfather”, commented Emily Hahn, who had ventured into the “East” and was living in China partly during the war and had been allured to the seductive power of opium smoking [5]. The Nationalist government and modernizing elites, who included increasingly powerful western trained medical scientists, blamed opium smoking for the weakness of China and the diseased body of the Chinese. They, as well as international pharmaceutical businesses, joined forces to promote penicillin, a modern medical drug, as a “cure” for China and Chinese illnesses.

Before penicillin, morphine, and heroin had been introduced by medical missionaries and modern medical institutions into China as opium “cures”, and the Anglo-American Tobacco company, supported by the Life Extension Institute in New York, distributed tens of millions of cigarettes in China for free, to wean the Chinese “from opium”. The former had helped to cultivate in the Chinese a taste for powerful synthetic drugs—“white pills” or “white powder” as they were called, where the white color came to symbolize “purity”, “cleanliness” and “better quality”—as well as alluring Chinese to the “healing power” of the syringe. American cigarettes succeeded in turning China into the world’s largest consumer of tobacco [6].

Soon after, in the autumn of 1944, Chinese scientists succeeded in producing their own penicillin, but they only managed to produce five bottles [7,8,9]. The amount was just enough for a dozen patients to last for a month. The amount was minimal compared to the much greater need. This meant, the bulk of penicillin continued to be imported from abroad, initially from the United States, and after 1949, from Europe. The limited supply and prohibitively high price meant that penicillin remained out of reach for the majority of people in China. On the other hand, unaffordability and unattainability made penicillin more desirable. As with opium, when it was first imported from overseas, it was as much a luxury as it was a medicine. Imported penicillin was seen by modern urban elites as a luxury, a badge of modernity, status, wealth, and power. Meanwhile, increasing numbers of urban poor, and an influx of war refugees from the countryside—a majority of whom were severely malnourished and exhausted and had not sufficient clothing nor housing to keep warm—suffered from tuberculosis and a wide range of other infectious diseases, such as whooping cough, diphtheria, and measles. Yet, many continued to resort to smoking cheap, low-grade illegal opium; sometimes taking heroin pills or injecting adulterated opiates to relieve symptoms such as pain and chills. Inevitably, some died of an overdose and many more were imprisoned [6]. Those who were lucky enough to be admitted into modern hospitals were treated with penicillin and survived [10,11]. Those who were treated, and their children, would always remember penicillin as life-saving. By the time the war drew to a close, the powerful allopathic medicine, penicillin, took over the medical use of opium in treating pneumonic infection, rheumatic fever, scarlet fever, and other diseases in China.

If in the second half of the 19th century, reliance on imported British opium had reduced China to a state of opium slavery, in the second half of the 20th century, a new western imported pharmaceutical, the powerful penicillin, would gradually replace opium as a medical panacea. This also marked the beginning of Chinese dependency on antibiotics. At the same time, as cigarettes from the United States took over the social function of opium, a huge number of Chinese suffered from resulting chronic obstructive pulmonary disease (COPD), a leading cause of mortality in China today. Along with biomass fuel use, cigarette/tobacco smoking has been identified as a key contributing factor to the high prevalence of COPD in the PRC [12]. Many suffering from COPD, rely on prescribed or over-the-counter antibiotics for treatment. However, the non-controlled use of antibiotics to treat COPD in clinical settings has often resulted in increased treatment failures, bacterial resistance, and even increased chances of relapse [13].

## 4. The Political Life of Antibiotics in Maoist China, 1949–1983

In the months immediately following the founding of the People’s Republic of China, in the autumn of 1949, an outbreak of bubonic plague devastated parts of northern China, surrounding the PRC’s new political capital, Beijing. Other parts of the country too were plagued by smallpox, meningitis, cholera, syphilis, and schistosomiasis. Official statistics, as presented by the new Minister of Health, Madame Li Dequan, in a government affair meeting in September 1950, identified that the mortality rate of infectious diseases was as high as 15 per thousand people, while 140 million people had fallen victim to infectious diseases annually [14]. Overwhelming poverty, disease, and poor health presented some of the biggest challenges for the new PRC government. Viewed as vital to productivity, military manpower, and national security, winning the battle against disease was considered to be of great political importance. Penicillin was utilized as a powerful weapon in the socialist war on infectious diseases and against so-called “feudal superstitious” or popular practices, to bring about a socialist cultural revolution. In multi-ethnic border regions of Hebei and Inner Mongolia as well as in the southwest, in particular, the Chinese Communist Party (CCP) sent doctors and health workers to treat local populations, who previously had little or no contact with allopathic medicine, with penicillin. The intervention produced immediate positive results. This outcome was used to demonstrate the efficacy of the new socialist medical and health care, and how Chairman Mao and the CCP were more powerful than Lord Guan and other local healing deities in guaranteeing people’s health and curing them of illnesses [15,16].

In the wake of the Korean War, the demand for penicillin would be amplified. As the supplies were running out and the PRC could no longer rely on the then enemy, the United States, for technical support (neither for production nor for import), an urgent need arose to practice self-reliance and mass produce penicillin domestically. In late 1953, the No. 3 pharmaceutical factory in Shanghai, under the leadership of Tong Cun and Ma Yucheng—two microbiologists trained in North America, began to pilot mass production of penicillin and achieved promising results. Both were honored by the state for their achievement [17].

As of 1958, partly driven by the political demand of the Great Leap Forward, which impelled the North China pharmaceutical factory to replace imported lactose with domestically produced glucose and starch, penicillin was mass-produced together with other newer antibiotics such as syntomycin, streptomycin, aureomycin, neomycin, tetracycline, erythrocin, and terramycin. The amount produced was, however, only sufficient to satisfy the restricted domestic supplies [17]. This meant that antibiotics were rationed by the state, as with most medical drugs and essential goods (including food as well as housing and other public services, such as free health and medical care). This rationing meant that penicillin and other antibiotics were a “privilege” granted to a select few individuals such as officials, health cadres, and higher rank doctors. Such selectivity would add to their desirability. On the other hand, those who were less privileged could find a way to access antibiotics by gaining the favor of those who controlled the drug. Those with power over antibiotics could in turn use them to gain favor for other services. Such a network of *guanxi*, or connection, came to eclipse money in importance in the PRC. It also made any effort of trying to restrict antibiotic use almost impossible, for refusal to grant access to antibiotics, even if for the sound reason of health risks, could potentially jeopardize one’s future chances of accessing other goods and services or opportunities for promotion.

The need for domestic mass production of antibiotics would be further exacerbated by the necessity to save foreign currencies for military spending, as well as to buy machines for rapid industrialization during the Great Leap Forward [18]. Simultaneously, the Great Famine put the already weak health system under serious threat, which collapsed in the rural countryside [16] (pp. 163–177). During the massive famine, to stave off hunger, a great number of villagers consumed soil and wild foods, such as leaves from the tree of heaven, as well as spoiled food or raw meats (including the flesh of human corpses) and fish. While the former caused severe constipation, many of the latter were toxic or harmful to human consumption. The toxicity of wild plants varied from mild to high. After consuming such a diet in large quantities and for long periods, many villagers suffered swelling and body pain. In some severe cases, bodies could no longer retain excess fluid, which then erupted through the skin, oozing out with yellowish color. In addition to widespread malnutrition, poisoning due to consuming food substitutes or spoiled foods was common in the countryside as well as in cities. In the absence of state medical and health care, people turned to local ritual healers and other types of folk medicine doctors or home remedies. Many of these relatively available and affordable undated remedies had unknown and often fatal side effects. The number of deaths from wild food poisons and from gastrointestinal bacterial infections after consuming spoiled foods was almost as high as the number of deaths due to starvation. In the aftermath of the famine, in some regions such as rural Shandong and Guizhou, the practice of opium smoking was reported to have revived, as were practices of other so-called feudal superstitions [16,19,20] (pp. 167–172).

It was during this crucial period that the PRC successfully obtained the technology and the licensing to mass produce semi-synthetic antibiotics, which resulted in a huge increase in the availability of antibiotics, thereby driving down the price. By August 1969, according to official reports, the price of more than 1230 kinds of patented medicines was slashed significantly, including a whole range of domestically produced antibiotics. The price of domestically produced penicillin, for instance, was 90 percent cheaper than in 1952, and 46 percent cheaper compared to one month before. Sulphanilamide tablets, too, became 63 percent cheaper than in 1952, and 13.3 percent cheaper than the previous month [21]. This radical reduction in the price of drugs, according to the *People’s Daily*, had reduced the health care burden of the masses living in mountainous, rural, and ethnic regions: it helped to “promote and consolidate the rural Co-operative Medical Service and hence the development of rural health care.” Following this banner news article came a news report about Barefoot doctors in rural Shanghai delivering “revolutionary” health care to villagers. The increased presence of the Barefoot doctors, as well as the increased affordability of western pharmaceuticals—with antibiotics showing the most significant price drop—were portrayed as victories for Mao’s revolutionary approach to health for the rural masses [21]. Thus, it became imperative for the Barefoot doctors to deliver antibiotics regardless of need or diagnosis.

The timing of this development is significant. The radical growth in the production of antibiotics in the PRC coincided with the advent of the Great Proletarian Culture Revolution which lasted ten years from 1966 to 1976, until after Mao’s death. As medical specialists lost their power during this period, the use of antibiotics became self-consciously divorced from the practice of western medicine. This “de-medicalization” of antibiotics allowed them to be easily assimilated into existing traditions. Barefoot doctors—the grassroots caregivers—were given the power to use antibiotics and other western medicines often without needing to adopt western medical theories of disease causality. Hence, it could be argued that unrestricted and unsupervised use of antibiotics by Barefoot doctors, endorsed and sanctioned by the state, helped create an antibiotic dependency in rural China and contributed to the “tragedy” of the overuse and misuse of antibiotics in the PRC. As Chen Zhiqian (more widely known in the West as C. C. Chen), the doyen of public health in modern China, rightly pointed out, the risk lies in the many Barefoot doctors—who were only given practical training in providing certain basic medical procedures and immunizations—but who were impelled to provide clinical work, from diagnosis to treatment, at the same level as a well-trained physician. Chen’s concern was confirmed by his daughter Chen Fujun’s field observation in rural Sichuan in the early 1970s, where she was most troubled by the problem of unrestricted and unsupervised use of antibiotics amongst the Barefoot doctors: “I witnessed many Barefoot doctors in rural Sichuan doing things they were not qualified to do. One thing I noticed was that they randomly gave out pills. They often randomly gave antibiotics to any villager who presented with a fever. This was because they did not know the cause [of the fever]. They were simply not trained to do diagnosis” [22,23].

The problem of the Barefoot doctors, the rural primary caregiver, over-prescribing or misusing antibiotics was by no means limited to rural Sichuan, nor was the problem short-lived. A 2012 joint study by researchers at Kunming Medical University in China and the Medical Faculty of Prince Songkla University in Thailand identified that in many parts of rural Yunnan in the southwest, the problem of over-prescribing antibiotics to pediatric patients was still widespread amongst rural primary caregivers, almost all of them former Barefoot doctors. In addition, the research also found that the level of supportive care was poor [24]. Undoubtedly, as Fujun pointed out, the Barefoot doctors’ inconsistent and inadequate training had contributed to their clinical incompetence and heightened both patient risk and their overreliance on antibiotics as a “cure-all”.

At the same time, the official promise of “Health Care for All” during this period saw the development of the Rural Co-operative Medical Service (RCMS), as advertised by slogans such as “It’s Free to See a Barefoot Doctor”, created a political demand and expectation for the Barefoot doctors to deliver therapeutic “cure”. This expectation was exacerbated by the propaganda that exaggerated the Barefoot doctors’ “infallible abilities” in curing many severe illnesses, including the removal of tumors. Burdened by these increased political demands and the heightened expectations, in addition to their lack of training and clinical incompetence, many Barefoot doctors, as well as other rural commune hospital doctors, resorted to prescribing the “magic bullet”—antibiotics. For instance, in Chonming Island, just outside of Shanghai, an official report in 1975 showed that primary care doctors at the Xinmin commune hospital prescribed antibiotics as an “insurance policy” to every village patient they saw regardless of their ailments [25].

In the PRC, the “de-medicalization” during the Long 1970s further increased the social and cultural functions of antibiotics, and thereby their use. Before the introduction of the RCMS, the majority of villagers in rural China had minimal or no contact with most western medicines. This was more so the case with antibiotics, due to their prohibitively high cost and limited availability before 1969. Even in the early years of the RCMS, antibiotics often remained a “luxury” reserved for cadres and those who had good relations with the cadres or the Barefoot doctors. As former Barefoot doctors recalled: “ordinary villagers could only get painkillers for free. If they wanted more expensive drugs, such as antibiotics, they would have to pay for it out of their own pocket.” “There was a shortage of drugs. Only those who had face (a good relationship with the cadres) had access to good and expensive drugs (like antibiotics)” [26]. Their inaccessibility and unaffordability made antibiotics that much more desirable when they became readily available, as they had acquired prestige and status associated with their use. By this point, the issue was not whether villagers’ preference was for western medicine or Chinese medicine. Rather, the “craving” for antibiotics was exacerbated by having little to no access to hospital care, due to distance, unavailability, or unaffordability. For many rural villagers, antibiotics became the more affordable and accessible alternative.

## 5. Medicine for the Poor in the Era of Marketization of Medical and Health Care, 1983–Present

The problem of the overuse of and over-reliance on antibiotics escalated with the post-Mao decentralization and marketization of health care. As of the early 1980s, the PRC government under Deng Xiaoping’s leadership abandoned the Maoist collective economic system in favor of neoliberal market capitalism—labelled as “socialism with a Chinese face”. After adopting the neoliberal market system, and with a large number of rural villagers migrating into coastal cities, the PRC government began to opt for a market model for financing health and medical services. Three years earlier in 1979, Kenneth Warren, the then Director of Health Science at the Rockefeller Foundation, and his colleague Julia Walsh favored “low-cost” but “high impact” targeted pharmaceutical interventions as an “interim” strategy for disease control in developing countries, and proposed the Selective Primary Health Care model [27]. The post-Mao government began to gradually abandon the Barefoot Doctor program, and moved away from the Maoist model that provided inexpensive and basic “comprehensive” primary health services and towards a selective primary health model with an emphasis on expertise defined as excellence, with specialized (tertiary) health care and a focus on short-term economic gain. The RCMS officially ended in 1983. Healthcare services were reduced or denied, following claims that they could not be sustained or afforded. Millions were disadvantaged in the process [28].

As the PRC government opted for a selective primary healthcare model, it also began to privatize providers. In the countryside, Barefoot doctors were reconfigured as village doctors and they opened private village clinics. To earn a living, as well as to sustain their services, they charged the patients fees. Those who could not afford the fees were often denied health care. Although authorities tried to remedy this by introducing measures to regulate prices for services, drugs, and equipment, this resulted in many providers focusing their activities on high-margin services by investing in high-tech equipment and expensive drugs. This meant health care became even more unaffordable for many villagers as the providers moved their efforts away from low-margin activities, such as day-to-day patient care. These factors radically changed the caregiver and care receiver relationship in the rural countryside. It meant more and more villagers, who had previously depended on their primary caregivers such as the Barefoot doctor, and respected their judgment, were forced to turn to relatively cheap western drugs, such as antibiotics, along with other equally cheap local indigenous remedies and other products, the latter often causing unknown and harmful side effects [23,29] p. 150. As one villager from Bingzhou in Shandong province reported in an interview: the “(advantage) nowadays is one can take pills if one gets ill” [30]. In multi-ethnic regions of Yunnan as well as in the Chinese researcher Yu Xiaoyan’s fieldwork of 2006 and 2009, as rural health and medical care entered the marketplace, traditional doctor and patient relationships shifted to a drug-centered system that almost always involved prescribing antibiotics. In an age when trust between the caregiver and care receiver became increasingly endangered, rural primary care doctors often prescribed antibiotics, not only to support themselves financially but also to maintain the trust of patients. Thereby, antibiotics began to take the credit for any cure, while doctors lost their caregiver function and essentially became drug dispensers. These changes often resulted in drug misuse and overuse that in turn further damaged doctor and patient relationships [29]. On the other hand, for some village doctors, biomedicine such as antibiotics, as well as patent Chinese medicines, was more convenient to obtain and use. Both types of medication could be found in regional pharmacies, and, while patent western and Chinese medicines were sometimes more expensive, they bore the imprimatur of “scientific” medical practice. Such changing perceptions have evolved over the past forty years with the erosion of traditional and local methods of healthcare practice and the increased prominence and knowledge of both western medicine and systemized TCM (Traditional Chinese Medicine), largely thanks to the PRC authorities’ concerted efforts to bring a “cultural revolution” to, and modernize, those “backward” multi-ethnic areas, such as Muchu’s village in the Himalayas. Until she was selected to be the village’s Barefoot doctor in 1973, Muchu, an interviewee for this research, reported that she and other villagers had no previous contact with allopathic medicine or Chinese medicine. During their initial Barefoot doctor training, she and fellow Barfoot doctors were taught “how to recognize Chinese medicinal herbs, as well as how to prepare and administer herbal medicines. Tibetan medicines normally come in grounded powder formula. We were not taught how to make them. We were taught how to make and administer Chinese herbal remedies or ready-made pills. We were encouraged to use the combination of western and Chinese medicine” [31]. Having been deprived of the knowledge of traditional Tibetan medicines, western and TCM filled the gap and became a preferred choice by local caregivers like Muchu and her patients. Restricted to a list of general drugs allowed, and available for them to use, they regularly resort to antibiotics for their perceived efficacy.

The overreliance on and overprescribing of antibiotics, however, is not limited to rural clinics. In urban cities, the unprecedented scale of urbanization further exacerbated the challenges to medical access and care quality, which in turn contributed to antibiotics misuse. In the early 1990s, in a drive to modernize China once more, the PRC government embarked on an epic-scale urbanization project. Millions of Chinese farmers were uprooted and displaced into coastal cities where employment in the new quasi-capitalist industries provided better economic incentives than the rural communities could provide. However, the rigid, state Hukou system of household registration meant rural migrants had little to no access to medical and health care and other urban welfare provisions, as a person’s local identity defined their access to the faltering state system of health care; in other words, one had to be treated where they were officially registered—a fact which has excluded millions from social welfare benefits. For the majority of migrant workers in China, antibiotics plus age-old home remedies were the only medical care to which they had access and could afford. Many migrant workers often lived in squalid conditions in cities with no access to clean water and did work that regularly exposed them to large amounts of toxins. As a result, pneumoconiosis persisted in being a major occupational disease in China. In 2019, according to data released by the National Health Commission, there were 15,898 new cases of pneumoconiosis, which accounts for more than 80% of the total occupational diseases for that year [32], and for which, prescribed and over-the-counter antibiotics, as well as available home remedies, were used in the continuing absence or inaccessibility of state medical care [33]. Many villagers and local health providers, interviewed between 2015 and 2018 across eight different provinces in China, reported that lack of access to promised state health care, as well as the low quality and the high cost of care, were the main reasons they continued to rely on relatively cheap and outdated remedies. Often adding to their “traditional” repertoire is the equally cheap, if not cheaper (partly thanks to China’s zero-profit drug policy) more widely available antibiotics [30]. Although in 2012, largely due to international pressure, the PRC authorities outlawed pharmacies and other private health providers from selling antibiotics to customers without a prescription, in general, patients can still walk into retail pharmacies and purchase antibiotics without a prescription as the majority of pharmacies in China do not require prescriptions, regardless of the official restriction [34,35]. What has changed, however, and has been adopted and is being enforced at the administrative level of township hospitals and above is that physicians may not prescribe antibiotics to more than 20% of their patients at any time.

In addition to human consumption, intensive farming practices accompanied by the overuse of antibiotics in animal husbandry and farming have also contributed to the occurrence and spread of AMR in the environment. Until 2014, more than half of the antibiotics consumed in China were administered to animals [36]. As in the United States and other western countries, the overuse of antibiotics in the agricultural sectors has made China’s AMR problem more severe. Although health authorities in the PRC introduced a series of policies and regulations to restrict and reduce human consumption of antibiotics, the use of antibiotics both for disease treatment and growth promotion in animals has remained unmonitored [37]. Ningxia province in northwestern China is often reputed for being less “polluted”, and food produced there is widely regarded by Chinese consumers as “safe”. A 2020 survey in Ningxia, however, shows that many of the small to medium-sized chicken farms there use antibiotics, and the majority of farmers reported that they had bought antibiotics without a prescription. However, they did not keep any records of antibiotic use. As with the problem of overprescribing antibiotics by rural primary caregivers, the problem of antibiotics overuse in animals seems to point to small farmers’ lack of formal training in agriculture and their perceptions of antibiotics as a panacea for all types of animal diseases. As the majority of farmers are relatively poor and unable to afford good standard hygiene and waste management, they use antibiotics for infection control or as prophylactics [38]. While the 2012 antibiotic regulation was introduced by the Ministry of Health, the responsibility for antibiotic use in animal feed and veterinary care is under the control of the Ministry of Agriculture and Rural Affairs (MOA). Coordination, or the lack of it between different government ministries, has always been an impending factor in crisis management and disease control and has made regulating antibiotics use in agricultural sectors difficult. It was only in 2019 that the MOA published regulations prohibiting the use of antimicrobial agents in animal feed and restricted the use of antibiotics in veterinary settings, which came into effect in 2020 [39,40].

In the early 1980s, the number of private drug manufacturers—many of which were illicit—mushroomed, and were linked to a new network of drug distribution in the countryside [41]. This development was partly driven by the instinct to be spared from the extreme poverty inflicted upon the people of China by the planned economy during the Mao era and partly due to the gradual economic liberalization introduced, first by local authorities, and eventually adopted by the national government of the PRC. Increased purchasing power and the easier accessibility created by this new network of production, distribution, and marketing have also contributed to the prevalence of antibiotic overuse.

Although in 2006, the PRC’s former Ministry of Health set up the Center for Antibacterial Surveillance (Mohcas) to address and monitor the growing problem of antibiotic misuse, as well as introduce various guidelines concerning antibiotic use and infection control, these guidelines have rarely been implemented, largely due to lack of local interest and resources. Meanwhile, antibiotics remained the most prescribed institutional medicine, and the issues of antibiotic resistance continued to increase [3]. As of 2009, the PRC authorities introduced a series of health reforms intending to enable the medical establishment to regain its public service role. These reforms allowed the Ministry of Health to launch a special campaign to enforce the rational use of antimicrobials in healthcare settings. This campaign achieved some success in hospitals in large cities and at the level of the township hospitals, where physicians may not prescribe antibiotics to more than 20% of their patients at any time, and, according to official statistics, drove the percentage of antibiotic prescriptions down from 68% to 58% for hospitalized patients and from 25% to 15% for outpatients [42]. However, consumer demand remains high and the problem of antibiotic overuse has persisted in both rural and urban settings across China.

However, even after PRC health authorities introduced a more rigorous regulation to prohibit pharmacies from the unrestricted selling of antibiotics in 2012, a high level of non-prescription access to antibiotics was still reported in different regions of China [34]. This was largely due to a lack of, or limited access to, hospital care for many. Notably, those living in less developed regions in the west and southwest have higher rates of self-medication, including antibiotics, compared to those living in more developed, larger cities where there are more hospitals [43,44,45]. Some rural residents reported stocking up on or consuming antibiotics prophylactically against colds [46]. If the desirability for antibiotics over the first three decades of the PRC was linked to the prestige derived from their use, the inability to access quality hospital care now counts as a major driver for the high-level consumer demand for antibiotics in the PRC [47,48]. Being able to purchase antibiotics with health insurance was also associated with higher rates of antibiotic self-medication [49,50]. Fear is another factor that has currently driven many to stockpile antibiotics. After the PRC authorities suddenly lifted all Covid restrictions in December 2022, cases of Covid infection increased drastically across China. In both urban and rural settings, as the already overloaded health system failed to cope with the rising demand, thousands have been denied critical medical care. An anxious population began to stockpile essential medicines: from fever-reducing medicines to cough relief remedies and antibiotics that were used to treat Covid related coughs and lung infections. In addition to the danger of antibiotic overuse, the stockpiling resulted in a shortage of antibiotics, as many pharmacies ran out of stock and smaller town hospitals and villager doctors could not stockpile antibiotics due to government restrictions. In the absence of antibiotics, some village doctors comforted sick patients with ginger soups [51].

## 6. Conclusions

This paper demonstrates that, since being introduced into China in the second half of the 20th century, following the introduction of penicillin, antibiotics, and their use have taken on a symbolic function of representing modern western medicine. Over time, this symbolic function separated itself from the system in which it was supposed to function. As in many other developing and developed countries, the remarkable growth of antibiotics in the People’s Republic of China reflected, on the one hand, the vital role they played in the treatment of bacterial infections, in particular in controlling widespread infectious diseases of bacterial origin, and, on the other hand, the weakness of a comprehensive national medical and health care system, together with a lack of professional controls and standards for antibacterial resistance. Although it is often the consumer that seems to determine the patterns of antibiotic use, there is no single factor or actor which could be blamed for the misuse or excessive use of antibiotics across China. As with opium during the early modern period, modern pharmaceuticals, such as penicillin and other antibiotics, entered a culture that had been accustomed to adapting new/foreign medicines and incorporating them into existing and diverse cultural and social systems. In the People’s Republic of China, there was a cascade of interrelated difficulties that accompanied the development and growth of antibiotics: the political demand for health improvement; the state-sponsored popularization of western medicine; the ongoing problem of lack of access to adequate medical and health care for the majority of the population; as well as the existing popular culture of self-medication. All worked in tandem to create the current dependence on antibiotics in China. In addition, the privatization and marketization of medical and health care in post-Mao China helped to build a new and ever-thriving network for the licit as well as illicit production, distribution, and marketing of antibiotics. This network has often proven difficult for the government to monitor. Additionally, easier accessibility and more affordability created by this new network, as well as the increased purchasing power of the consumer further contributed to the prevalence of antibiotic overuse in the late 20th and early 21st century.

Distrust of the state’s primary care system has also resulted in increasing rates of self-medication; adding to people’s “traditional” repertoire with the equally cheap and widely available antibiotics. Although in 2012, the Ministry of Health of the PRC introduced a regulation to prohibit pharmacies from the unrestricted selling of antibiotics, one could still obtain antibiotics without a prescription, especially in small cities and towns where local authorities often treat government policies as mere guidelines and don’t feel obligated to implement them due to a variety of local factors. In addition, there is little sign of declining consumer demand for antibiotics. There have been reports of physicians and pharmacists preparing cocktails of traditional Chinese medicines mixed with antibiotic agents for patients with upper respiratory tract infections (URTI) [46,52]. However, the issue of human overconsumption of antibiotics is worsened by antibiotic overuse in the agricultural sector; largely driven by the consumer demand for the production of high-protein foods, due to a rise in living standards that accompanied the post-Mao economic reforms. Exposure to excess consumption of antibiotics through food sources has only exacerbated antibiotic resistance. One further potential danger is, as with opium prohibition in the late 19th and early 20th century, once demand and supply have been established, official restrictions can prove counterproductive driving antibiotic selling and consumption underground and creating further problems that may be equally or even more catastrophic than drug resistance.

## Data Availability

The data used and/or analyzed during the current study are available from the author on reasonable request.

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
