# Peer review of "Antibiotic Culture: A History of Antibiotic Use in the Second Half of the 20th and Early 21st Century in the People’s Republic of China"

_antibiotics, 2023, doi:10.3390/antibiotics12030510_

Round 1

Reviewer 1 Report

An interesting text that deserves a large readership. One learns a lot from it with regard to the development of the medical system and its acceptance and perception in the PRC. The analysis is convincing  without being overcritical. Well done.

Author Response

Thank you for the helpful comments

Reviewer 2 Report

Interesting article on the history of antibiotic use in the Republic of China.

The review satisfactorily describes the political, economic, social and medical reasons for the misuse of antibiotics during half of the 20th and 21st centuries.

The introduction and objective of the study is missing

Important aspects of social, political, and cultural aspects of historical behavior are explained, culminating with the reason for the irrational use of antibiotics. However, I do not know if it falls within the scope of the magazine

2.   It is important to know the history and evolution of the use of antibiotics, the biggest disadvantage is the limited bibliography used, and that the existing one is usually anecdotal.

3.       What does it add to the subject area compared with other published material?

The history and evolution of the use of antibiotics and know the causes that led to the misuse of antibiotics.

4.       What specific improvements should the authors consider regarding the methodology? What further controls should be considered?

Add the introduction, add the search criteria and align to (SANRA—a scale for the quality assessment of narrative review articles by Christopher Baethge et al) DOI: https://doi.org/10.1186/s41073-019-0064-8

 5.       Are the conclusions consistent with the evidence and arguments presented and do they address the main question posed?

The revision does not have an explicit objective, it is necessary to add

6.       Are the references appropriate?

Most of the bibliography used contains historical aspects but few of them are verifiable scientific data, rather they are anecdotal.

Well done

Author Response

Thank you for the comments. I have made some revisions accordingly, e.g. to add an Introduction which states clearly what the Objectives of my study are.  I also revised the Conclusion. As for the other point you raised with regard to the methodology, I disagree as this study is not a literature /narrative review nor an evaluation of official policies regarding to AMR. It is a historical study exploring different historical factors contributed to the AMR in the PRC. 

Reviewer 3 Report

This article has significance for readers to understand the history of antibiotic use in China. Here are a few comments and recommendations.

The author did not mention much about the policies on the sale and use of antibiotics after 2000 (except for the policies mentioned several times in 2012), so the explanation of the purchase and use of antibiotics was somewhat general. Recent years, antibiotics were basically prescribed by physician, although walking into a local pharmacy and buy antibiotics over the counter still existed, so I think the author's statements in these 349-354 lines are not so rigorous. It is better to supplement more policies after 2000, including after 2012.

The previous parts of this article were all about the human use antibiotics, and the last paragraph of agricultural sector was suddenly inserted (including 428-430 in the conclusion), which was a bit abrupt. If you want to talk about agriculture sector, you need to add more space, or you don't have to talk about it at all, then the title is limited to the history of human use of antibiotics.

Parts 1,2, and 3 can be subtitled according to the timeline of events. It is better to list several points in conclusion as well.

The author mentions or quotes interviews from his previous research (269,310, etc.) at various points in the article, it is better to provide more background information.

It might mean “the Operation Ichi-Go” in the last words of line 28 not “Operation Ichi-Gothe”.

Author Response

Thank you for your helpful comments. I have revised the article accordingly.